# Ethnomedicinal Use, Phytochemistry, Pharmacology, and Toxicology of *Euphorbia resinifera* O. Berg. (B): A Review

Oumaima Hmidouche [1,2], Khadija Bouftini [1,2], Abdelbasset Chafik [3,4,5], Sara Khouri [6], Halima Rchid [2,7], Abdessadek Rahimi [1,8], Mostafa Mimouni [8], Elbekay Maarouf [8], Fatna Zaakour [9], Rachid Nmila [2,10] and Aya Khouchlaa [1,*]

1   Laboratory of Biochemistry, National Agency of Medicinal and Aromatic Plants, BP 159, Taounate 34000, Morocco
2   Laboratory of Biotechnology and Valorization of Plant Resources, Faculty of Sciences, Chouaib Doukkali University, El Jadida 24000, Morocco
3   Higher School of Technology of El Kelâa des Sraghna, Cadi Ayyad University, Beni Mellal Road Km 8 BP 104, El Kelâa des Sraghna 43000, Morocco
4   Laboratory of Biochemistry, Neurosciences, Natural Resources and Environment, Faculty of Sciences and Technologies, Hassan First University, Settat 26000, Morocco
5   Bioresources and Food Safety Laboratory, Faculty of Science and Technology, Cadi Ayyad University, Boulevard Abdelkrim Khattabi, BP 549, Marrakech 40000, Morocco
6   Public Hospital Establishment (EPH), Graine Amar Ayachi El Hadjar, Annaba 23004, Algeria
7   Multidisciplinary Faculty Sidi Bennour, Chouaib Doukkali University, Av. Jabrane Khalil Jabrane BP 299, El Jadida 24000, Morocco
8   Laboratory of Applied Chemistry and Environment (LCAE), Faculty of Sciences Oujda (FSO), University Mohammed First (UMP), Oujda 6000, Morocco
9   Laboratory of Sustainable Agriculture Management, Department of Agricultural and Environmental Engineering, Higher School of Technology Sidi Bennour, Chouaib Doukkali University, Av. des Facultés, El Jadida 24123, Morocco
10  Higher School of Education and Training, Chouaib Doukkali University, El Jadida 24000, Morocco
*   Correspondence: aya.khouchlaa@gmail.com

**Abstract:** *Euphorbia resinifera* (Zaggûm or Tikiut) is an endemic species of Morocco that grows in the Middle Atlas Mountain range. It is used in folk medicine to treat several diseases, especially various types of cancer. Aim of the review: In this review, we critically highlighted and discussed previous reports on *E. resinifera*, concerning its botanical description, taxonomy, geographical distribution, and medicinal use. In addition, bioactive compounds, toxicology, and pharmacological effects were reported. Materials and methods: We searched various scientific databases, such as Scopus, PubMed, Web of Science, SpringerLink, SciFinder, Wiley Online, and Google Scholar, to collect data on *E. resinifera*. Studies involving *E. resinifera* or its bioactive compounds with regards to antitumor, anti-inflammatory, antileishmanial, antiprotease, immunomodulatory, irritant, and lysosomal activities are discussed here. Results: *E. resinifera* has been widely used in folk medicine practice in Morocco to treat several diseases such as diabetes, cancer, and inflammatory skin conditions, as well as to heal wounds. In phytochemistry studies, biochemical compounds have been identified from *E. resinifera* belonging specially to terpenoids. Indeed, in vitro and in vivo pharmacological investigations showed that extracts and compounds from the latex of *E. resinifera* exhibited a wide spectrum of biological properties, particularly antioxidant, antimicrobial, antitumor, anti-inflammatory, antileishmanial, and immunomodulatory and neuroprotective activities. Conclusion: The use of *E. resinifera* in conventional medicine is supported by processes founded on biological evidence. However, in-depth research is necessary to prove the safety and efficacy of *E. resinifera* latex extracts and their compounds and to clarify their pharmacological mechanisms. In addition, pharmacokinetic and pharmacodynamics studies are required for extracts and bioactive compounds.

**Keywords:** *Euphorbia resinifera*; ethnomedicinal use; phytochemistry compounds; biological activities

## 1. Introduction

*Euphorbia resinifera* O. Berg. & C.F. Schmidt belongs to the *Euphorbiaceae* family and the *Euphorbia* genus. *E. resinifera* is an endemic species of Morocco and it is distributed in the center of the country, in the Middle Atlas Mountain range in the regions of Azilal and Beni Mellal [1]. This species is a perennial leafless cactus-like plant (60 cm in length) [2] which presents four-sided, almost square stems with short brown spines [3]. *E. resinifera* is known locally in Morocco as "Zaggûm" in Arabic and "Tikiut" in Berber. The spurge resin is known as "Ferbyûn" or "Luban al-magribî" [4].

In traditional medicine, *E. resinifera* is cited in various ethnobotanical studies and used traditionally against different illnesses, especially to treat cancer [5–11]. It is also used to treat diabetes [12,13] and hypoglycemia [9]. This species is also used for healing wounds [14], healing venomous stings [4,5,8], as a hair tonic and for hair care [7,15], and is also prescribed to terminate a pregnancy [4,12], as a laxative, and as an anti-inflammatory agent [9]. *E. resinifera* is traditionally used against different illnesses, including respiratory diseases (flu, asthma, allergies, etc.), circulatory disorders, metabolic disorders, digestive diseases, diseases of the reproductive system, headache, weakness, yellowing, and intoxication [8].

Many researchers have reported the pharmacological activity of *E. resinifera* extracts including antioxidant [3,16,17] antitumoral [18], antibacterial [1,3,17–19], antifungal [3], antitrypanosomal [20], neuroprotective [21], against irritation and skin damage [22,23], and antileishmanial effects [20]. In addition, other studies investigated the enzymatic activities of xanthine oxidase, lipoxygenase, acetylcholinesterase, and tyrosinase [17], its liver effects [24], lysosome biosynthesis [25], and clinical trials [26–30]. However, to the best of our knowledge, the in vivo toxicological effects of *E. resinifera* extracts have been rarely studied [31]. The presence of several chemical classes in *E. resinifera* extracts have been identified using phytochemical investigation. Terpenoids represent the dominant phytocompounds class in the latex extracts.

However, several studies affirmed the pharmacological properties of *E. resinifera* and its compounds. This review was published to summarize all these studies of this Moroccan endemic plant, *E. resinifera*. This promotes us to write the current review highlighting different aspects of *E. resinifera* including taxonomy, botanical description, distribution, ethnobotanical investigations, phytochemistry, and all pharmacological reports.

## 2. Materials and Methods

All data concerning taxonomy, botanical description, distribution, ethnobotanical investigations, phytochemistry, and pharmacological activities of *E. resinifera* were collected, analyzed, and summarized from the published literature with the use of different sources of scientific search engines such as SpringerLink, SciFinder, Scopus, Web of Science, PubMed, Google Scholar, ScienceDirect, and Wiley Online. For this bibliometric investigation, several terms were used such us "*E. resinifera*", "ethnomedicinal of *E. resinifera*", "*E. resinifera* essential oil", "antioxidant activity of *E. resinifera*", "antidiabetic activity of *E. resinifera*", "antibacterial activity of *E. resinifera*", "antileishmanial activity of *E. resinifera*", "activities of *E. resinifera*", and "the chemical composition of *E. resinifera*". Using the PubChem database, the IUPAC names of the biological compounds were verified. Using ChemDraw Ultra 12.0 software, the chemical structures were drawn.

## 3. Results and Discussion

### 3.1. Botanical Description

The Euphorbiaceae family contain 8000 species, usually plants with latex [32], including *E. resinifera*. This last is a succulent shrub, up to 60 cm in height and can reach 1–1.5 m in original. As presented in Figure 1, the stems are green, and four-sided almost square, forming multi-stemmed cushion-shaped clumps up to 2 m wide. Several small leaves are rarely seen and drop in the spring [3]. The spines are short and stipular (2–10 mm), on a well-defined oval escutcheon; a gemmiparous point is located a little below the middle

of the interstipular space. The size of seeds is 2.7–3 mm and the capsule about 4 mm or more wide. The style is 1 mm long [32]. *E. resinifera* flowering starts in spring and appears at the end of the stems. The fruit of this plant is a small trilobed capsule, trilocular (three carpels) [33]. The pith and cortex of *E. resinifera* contain long branching, laticiferous cells (Figure 2). When this plant is wounded, the latex is in the form of milky drops [34].

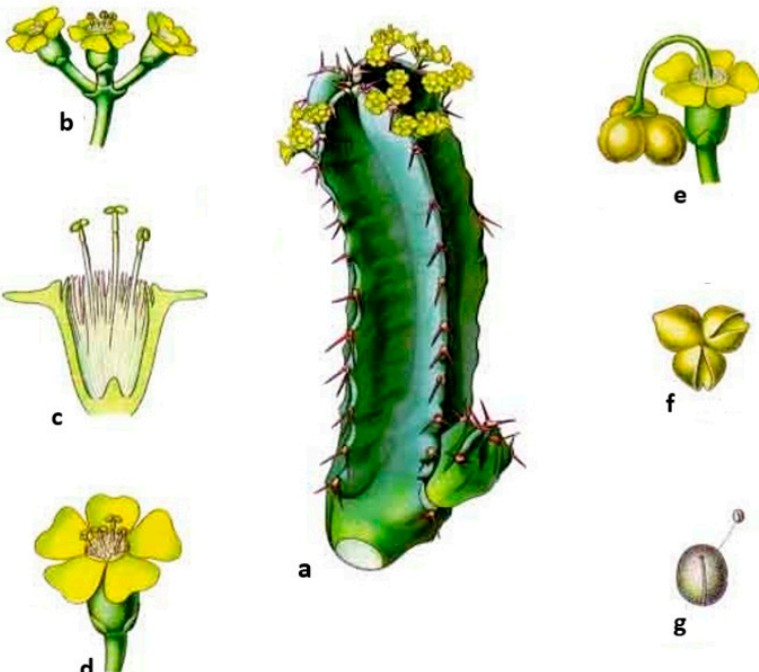

**Figure 1.** *E. resinifera*: (**a**) flowering branch, (**b**) head inflorescence, (**c,d**) staminate, (**e**) pistillate flower, (**f**) fruit, and (**g**) seed [34].

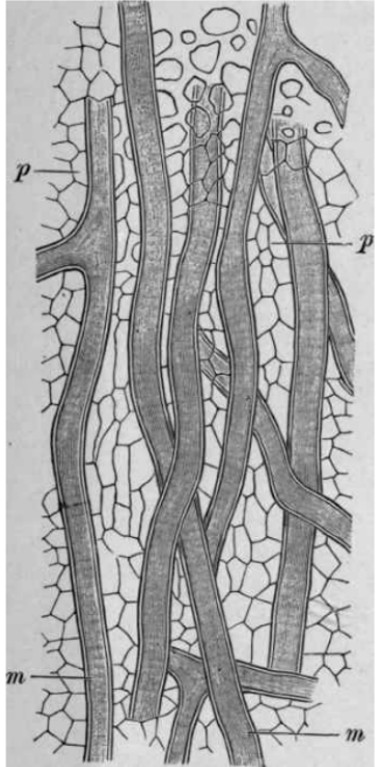

**Figure 2.** Laticiferous cells in the stem of *E. resinifera*. (*p*) parenchyma, (*m*) laticiferous cells [34].

### 3.2. Taxonomy and Geographical Distribution

*E. resinifera*, also named Euphorbia resin, is a dicotyledon species belonging to the *Euphorbiaceae* family [35]. This family, made up of more than 300 genera and 8000 species, is one of the most important families of medicinal plants [36,37], being one of the largest families and the most cosmopolitan of the under-branching angiosperms [38]. The genus *Euphorbia* includes more than 2000 species and constitutes the largest genera of flowering plants which contain Euphorbium, an irritant latex [39].

*E. resinifera*, a Moroccan endemic species, is located mainly in the center of the country (regions of Azilal and Beni Mellal, Middle Atlas Mountain range) [1]. It extends from Demnate in the Central High Atlas in the southwest to the Elksiba area in the Middle Atlas in the northeast, over an area of 316,500 ha, but appears as scattered infiltrations from Demnate to Ait Ourir (center of the High Atlas) due to a complexity of factors linked to the characteristics of the soil, the paleogeography and the exacerbated climate. It extends into the central valley of the High Atlas in small, isolated units. This could explain why other ecological requirements must be taken into account in terms of the distribution of this endemic species in the context of climate change. From a bioclimatic point of view, the spurge zone is found in arid to subhumid bioclimates but includes temperate, cool, and exceptionally cold variants [36,40,41].

### 3.3. Population Status

In the areas of Azilal and Beni Mellal, *E. resinifera* grows in the rocky mountain at altitudes between 600 and 1500 m where the summer is hot and the winter is cold. Recently, this species was found at altitudes higher than 1900 m [40]. Indeed, with such climatic conditions, agriculture is not feasible. In fact, *E. resinifera* plays a significant role in the economy of the areas of Azilal and Beni Mellal, where the population of *E. resinifera* produces a honey of high nutritional and therapeutic quality. These things considered, the monofloral *Euphorbia* honey of these areas was labeled as Protected Geographical Indication (PGI) [36]. Altogether, the geographical distribution and economic value of the population of *E. resinifera* in the areas of Azilal and Beni Mellal lead to the classification of these areas as some of the most valuable territories for honey production in the Mediterranean area [42].

### 3.4. Possible Threats

*E. resinifera* is under great anthropogenic pressure, because it is a local development economy in the mentioned areas [36]. In addition, as mentioned above, *E. resinifera* was found at altitudes higher than 1900 m, which indicates that this species has an adaptive strategy that helps it survive and grow in certain environments; this is a good indication in light of current climatic changes [40]. With this in mind, *E. resinifera* is a species that could be considered as not threatened.

### 3.5. Conservation

*E. resinifera* is locally well conserved; the milky latex or sap it contains, which causes intense inflammation of the skin and eye, keeps it safe from human activities [43]. However, emergency measures should be taken to protect this wild and endemic species in Morocco.

### 3.6. Ethnomedicinal Use

Several ethnobotanical studies have reported the importance of *E. resinifera* in traditional folk medicine. The applications of different parts of *E. resinifera* in traditional global systems are listed in Table 1. Different parts of this plant have been used traditionally to treat numerous illnesses. The traditional use of *E. resinifera* depends on the part used of this plant.

In Morocco, treating numerous types of cancer by *E. resinifera* has been reported as a famous prescription. Different regions of Morocco use this plant as a remedy for several types of cancer. The use of this species to heal cancer by the population of North Atlas of Azilal (Morocco) has been investigated by an ethnomedicinal study [7]. The entire plant (dry

or fresh) has been widely used as an antitumor [9]. In 2019, Samouh et al. [11] highlighted the use of *E. resinifera* against cancer by patients in the unit of oncology (University teaching hospital Ibn Rochd of Casablanca, Morocco) and reported the use of resin naturally to treat breast, colon, lung, uterus, and otorhinolaryngology (ORL) cancers, leukemia, and stomatology. Resin, roots, leaves, stem, and bark have been used in decoction form for their medicinal role in the Casablanca region (Morocco) for treatment of cancer. The populations of the Atlas Mountains of Azilal and Beni Mellal have used orally the stems (without latex or bark) for the treatment of breast cancer and female genital tract cancer [8]. Grinding of the aerial parts mixed with honey (or honey alone) has been used orally against different types of cancer [8,10]. A small drop of latex has been used by the same population to treat skin cancer for external use [8]. Moreover, the ground aerial parts mixed with honey are used to treat cancers by the population of Rabat [8].

In addition to their antitumoral use, several researchers reported the use of *E. resinifera* stems, leaves, fruits, flowers, and latex in the treatment of diabetes. The population of the Central High Atlas Mountain of Morocco, center of the Atlas chain of Azilal North, mountain chains of the Middle Atlas, Ifrane, Khenifra, Ouarzazate, Tinghir South, Beni Mellal, and the population of Marrakech used numerous parts of *E. resinifera* against diabetes [5,9,12,13,44].

*E. resinifera* has been also used traditionally for cosmetic purposes. The population of the Middle Atlas, Ifrane, and Khenifra Mountain ranges have used this species (dry and fresh whole plant) as a laxative and for cosmetic purposes [9]. The population of Ksar Lakbir (Northern Morocco) has used *E. resinifera* for hair tonic and hair care [45]. Another ethnobotanical study in the North of the Atlas of Azilal confirmed the cosmetic and therapeutic effects of *E. resinifera* on the hair [7].

The use of *E. resinifera* to treat teeth, inflammation, and poisonous bites has also been reported. In the Northern Azilal located in the center of the Atlas Mountains, El Alami et al. proved its use for the treatment of teeth [7]. This was confirmed, four years later, in the study of El Alami et al. [8] conducted in the population of the Atlas Mountains of Azilal and Beni Mellal. In this study, a small drop of latex from *E. resinifera* was used in traditional medicine against toothache. Other ethnobotanical studies of this plant reported the used of *E. resinifera* in folk medicine in the treatment of inflammation. In the regions of the Middle Atlas Mountain ranges, Ifrane, Khenifra, the whole plant (dry and fresh) is renowned for its anti-inflammatory effect [9]. In the north of Azilal, located in the center of the Atlas Mountains, and in Beni Mellal, located in the northwestern region of the Central High Atlas, the population has used warm water mixed with honey and latex to cure skin inflammation [7,8]. In addition to their anti-inflammatory effect, several parts of *E. resinifera* have been used against poisonous bites. The stems, fruits, and flowers of *E. resinifera* are used against venomous bites by the population of the Central High Atlas of Morocco [5]. Moreover, latex of *E. resinifera*, mixed with hot water and honey, has been used for the same reason (snakebites and scorpion stings) by the population of the Atlas Mountains of Azilal and Beni Mellal [8].

Depending on the geographical region, *E. resinifera* has been used traditionally against many diseases. The people of the Atlas Mountains, particularly those of the Azilal and Beni Mellal regions, have used lukewarm water mixed with latex and honey against poisoning. Mixed with honey, the powder of dried areal parts has been used to treat goiter. The small drop of latex for warts, stem (without latex or bark), and milk juice have been used for the treatment of female genital cysts. Seed powder, mixed with honey, was reported to treat digestive disorders [8]. Other therapeutic effects have been reported by the population of the Atlas Mountains of Azilal and Beni Mellal. In this study, this population used *E. resinifera* mixed with honey to treat respiratory diseases, circulatory disorders, metabolic disorders, digestive diseases, diseases of the body, diseases of the reproductive system, headaches, weakness, yellowness, angina, rheumatism, and cysts [8].

According to those researchers, it is confirmed that *E. resinifera* is used in traditional medicine by different populations of several Moroccan regions to treat many diseases,

especially cancer. Despite the main disorders cited with the use of *E. resinifera* in all these ethnobotanical research studies, this traditional medicine requires a deep analysis of pharmacological activities, bioactive compounds, and toxicological studies.

**Table 1.** Ethnomedicinal use of *E. resinifera*.

| Area of Study | Used Composition/Part | Mode of Preparation | Traditional Use | References |
|---|---|---|---|---|
| Central High Atlas of Morocco | Stems Fruits Flowers | Poultice Infusion Powder | Venomous stings Against paralysis (kneaded with flour or semolina and egg white) Abortifacient Antidiabetic | [5] |
| Tinghir South, Ouarzazate, and Center of the atlas chain of Azilal North | Flowers, latex | Infusion, latex in water | Antidiabetic | [12] |
| Marrakech | Dried plants, euphorbia honey | Not reported | Antidiabetic | [13] |
| Mountain chains of the Middle Atlas, Ifrane National Park, and Khenifra National Park | Entire plant (dry and fresh) | Not reported | Laxative Anti-inflammatory Hypoglycemic Antitumor activity Cosmetic purposes | [9] |
| Greater Casablanca, Morocco | Roots, leaves, stem, and bark | Decoction | Cancer | [6] |
| The Northern Azilal located in the center of the Atlas Mountains (Morocco) | Not reported | Not reported | Inflammation Skin Tumor Teeth treatment Skin care and hair | [7] |
| Azilal and Beni Mellal located in Atlas Mountains (Morocco) | Warm water mixed with latex, and honey | External use | Skin inflammation | [8] |
| | | Oral | Intoxications | |
| | | External use and/or oral | Snakebites and scorpion stings | |
| | Aerial parts (powder) with honey | Oral | Goiter | |
| | Latex | External use | Skin cancers Warts Toothache | |
| | Stem (without latex or bark) and milk juice | Oral | Cysts of the female genital tract Female genital tract cancer Breast cancer | |
| | Ground aerial parts mixed with honey | Oral | Cancers | |
| | Seeds (powder) with honey | Oral | Digestive issues | |

**Table 1.** *Cont.*

| Area of Study | Used Composition/Part | Mode of Preparation | Traditional Use | References |
|---|---|---|---|---|
| Azilal and Beni Mellal located in Atlas Mountains (Morocco) | Honey | Oral | Respiratory diseases (flu, asthma, allergies, etc.) Circulatory disorder Metabolic disorders Digestive diseases Diseases of the reproductive system Headache Weakness and yellowing Cancers Angina | [8] |
| | Honey | External use | Skin conditions | |
| Rabat | Aerial parts | Ground with honey (oral) | Cancers | [10] |
| Ksar Lakbir district | Seeds | Mixed with *Lawsonia inermis* and kneaded with water | Hair care | [45] |
| | | Melted in olive oil | Hair tonic | |
| Casablanca | Resin | Natural | Cancers (breast, colon, lung, uterus, ORL, leukemia, stomatology) | [11] |
| Beni Mellal | Leaf stems | Decoction | Antidiabetic | [44] |

### 3.7. Phytochemistry

Several studies have shown that extracts of *E. resinifera*, especially latex extract, contain many chemical compositions [16,21,25,46–61]. Table 2 summarizes the class of bioactive compounds, the plant organ, the country of origin, and the most abundant compounds of *E. resinifera*, namely terpenoids. Terpenoids were the main class identified.

**Table 2.** Phytochemistry of *E. resinifera*.

| Used Part/Country | Extract | Compound Groups | Main Compound | References |
|---|---|---|---|---|
| - (Spain) | Latex extract | Terpenoids | 12-Deoxy-16 hydroxyphorbol 13,16-diesters | [21] |
| - (Morocco) | Latex extract | Terpenoids | Euphorbioside | [62] |
| Stem (Italy) | Latex extract | Terpenoids | Euphorbioside A Euphorbioside B | [47] |
| Aerial part (France) | Latex extract | Terpenoids | Ingenol (16.71 µg/mL) | [48] |
| Aerial parts (Morocco) | Latex extract | Terpenoids | Resiniferatoxin Proresiniferatoxin Resiniferonol 12-deoxy-phorbol-13-angelate 12-deoxy-phorbol-13-isobutyrate | [49] |
| - (Morocco) | Latex extract | Terpenoids | Ingenol-3-acylates 12-deoxyphorbol-13-ester-20-acetates | [50] |
| - (China) | Latex extract | Terpenoids | 3β-hydroxy-12α-methoxy-24-methylene-lanost-7,9(11)-dien; 3β-hydroxy-12α-methoxylanosta-7,9(11),24-triene; 3,7-dioxo-lanosta-8,24-diene; 3,7-dioxo-24-methylene-lanost-8-en | [51] |

**Table 2.** *Cont.*

| Used Part/Country | Extract | Compound Groups | Main Compound | References |
|---|---|---|---|---|
| - (China) | Ethanol extract | Terpenoids | Cycloartan-1,24-diene-3-ol; Cycloartan-1,24-diene-3-one; 3β-hydroxy-lanosta-8,24-diene-11-one; Eupha-8,24-diene-3β-ol-7,11-dione; Eupha-8,24-diene-3β,11β-diol-7-one; Eupha-24-methylene-8-ene-3β-ol-7,11-dione; | [51] |
| - (China) | Latex (methanol extract) | Terpenoids | Euphatexol C Euphatexol D Euphatexol E Euphatexol F Euphatexol G | [52] |
| - (Morocco) | Latex extract | Terpenoids | Euphol Euphorbol | [54] |
| - (Morocco) | Latex extract | Terpenoids | Lanosta-8,24-dien-3b-ol; (3S,5S,10S,13S,14S,17S)3b-acetyl-25,26,27-trisnorlanost-8-en-24-oate | [53] |
| Stem (USA) | Latex extract | - | Phorbic Acid | [56] |
| Stem (USA) | Latex extract | Terpenoids | Glycoside Lipid | |
| Stem (Morocco) | Latex extract | Terpenoids | Euphorbioside C | [56] |
| - (China) | Latex Extract | Terpenoids | Euphatexol A Euphatexol B | [57] |
| Aerial parts (Morocco) | Hexane extract | Hexane compounds | Ethyl linoleate (13%) 1,3,4-Trimethyl-3-cyclohexanyl-1-carboxaldehyde (14%) Heptacosane (26%) | [18] |
| Aerial parts (Morocco) | Dichloromethane extract | Dichloromethane compounds | 1,3,4-Trimethyl-3-cyclohexanyl-1-carboxaldehyde Ledane (13%) 1,4-bis-(2′-cyclopropyl-2′-methylcyclopropyl)-but-2-en-1-one (5%) Cis-Z-α-Bisabolene epoxide (7%) | [18] |
| Aerial parts (Morocco) | Methanolic extract | Methanolic compounds | Cis-Z-α-Bisabolene epoxide (5%) Methyl arachidonate (9%) Methyl ester 9,11-(1,1′-bicyclopropyl)-octanoic acid (4%) | [18] |
| - (China) | Methanolic extract | Terpenoids | Euphorols A-I | [59] |
| - (China) | Methanolic extract | Terpenoids | Iso-maticadienediol; 3β-hydroxy-25,26,27-trinor eupha-8-ene-24-oate; Dammarendiol II; 25,26,27-trinorTirucall-8-ene-3β-ol-4-acid; eupha-8,24-diene-3-ol-26-al Inonotusane C; Eupha-8,24-diene-3β-ol-7,11-dione Inoterpenes A-B; Eupha-24-methylene-8-ene-3β-ol-7,11-dione | [63] |

**Table 2.** *Cont.*

| Used Part/Country | Extract | Compound Groups | Main Compound | References |
|---|---|---|---|---|
| - (China) | Methanolic extract | Terpenoids | Euphorol K<br>Euphorol J<br>Kansuinone | [60] |
| - (China) | Methanolic extract | Terpenoids | Euphoresins A<br>Euphoresins B | [61] |
| - (China) | Methanolic extract | Terpenoids | Euphorblin A<br>Euphorblin B<br>Euphorblin C<br>Euphorblin D<br>Euphorblin E<br>Euphorblin F<br>Euphorblin G<br>Euphorblin H<br>Euphorblin I<br>Euphorblin J<br>Euphorblin K<br>Euphorblin L<br>Euphorblin M<br>Euphorblin N<br>Euphorblin O<br>Euphorblin P<br>Euphorblin Q | [64] |
| - (China) | Methanolic extract | Terpenoids | Euphorbiumrin A<br>Euphorbiumrin B<br>Euphorbiumrin C<br>Euphorbiumrin D<br>Euphorbiumrin E<br>Euphorbiumrin F<br>Euphorbiumrin G<br>Euphorbiumrin H<br>Euphorbiumrin I<br>Euphorbiumrin J | [64] |

-: not reported.

### 3.7.1. Terpenoids

Previous studies investigated the chemical components of *E. resinifera* extract and led to an identification of a wide range of terpenoids (Figure 3). The majority of terpenoids were identified in the latex of this plant. The first study was conducted, in 1969, by Nordal and Benson (1969), who showed the presence of two types of terpenoids (glycoside and lipid) in the latex of the stem (from Oslo, Norway) obtained by a series of extractions (95% boiling ethanol and after boiling water) [56]. Hergenhahn et al. (1975) isolated five terpenoids from *E. resinifera* dried methanolic extract of fresh latex (from Rabat, Morocco) (resiniferatoxin, proresiniferatoxin, resiniferonol, 12-deoxy-phorbol-13-isobutyrate, and 12-deoxy-phorbol-13-angelate) [49]. In 1984, Hergenhahn et al. (1984a) isolated two other terpenes from *E. resinifera* latex collected from Rabat (Morocco): Ingenol-3-acylates and 12-deoxyphorbol-13-ester-20-acetates [50]. Ingenol was identified to be the richest compound in the latex extract from the aerial part of *E. resinifera* with a value of 16.71 μg/mL using an HPLC method [48]. In 2002, the composition of *E. resinifera* latex (Italy) using NMR

was determined by Fattorusso et al. [47]. They identified two terpenoids: euphorbioside A, and euphorbioside B. In *E. resinifera* latex, lanosta-8,24-dien-3β-ol was the main triterpene isolated [55]. Seven years later, Mallon et al. were able to determine the presence of the two terpenoids, euphol and euphorbol, using the LC-MS instrument [54]. Within the same year, euphorbioside had been identified by Farah and Echchahad, et al. from *E. resinifera* latex harvested from Demnat (Morocco) using an HPLC method [59]. In 2016, Wang and coworkers were able to isolate several triterpenes from the methanolic extract of Euphorbium (from Urumqi, China) using the 2D NMR, IR, UV, 1D, and HR-ESI-MS methods. They identified and isolated nine novel compounds: three tirucallane triterpenes (Euphorol E, F, and G), and six euphane triterpenes (Euphorol A, B, C, D, H, and I) [59]. Two years later, Wang and investigators isolated 10 triterpenes compounds (3β-hydroxy-25,26,27-trinor eupha-8-ene-24-oate, iso-maticadienediol, 25,26,27-trinorTirucall-8-ene-3β-ol-4-acid, dammarendiol II, eupha-8,24-diene-3-ol-26-al, lnonotusane C, eupha-8,24-diene-3β-ol-7,11-dione, inoterpene A, inoterpene B, and eupha-24-methylene-8-ene-3β-ol-7,11-dione) from the methanol extract of *E. resinifera* latex using a variety of chromatographic methods including semi-preparative HPLC, silica gel, and ODS. They identified their structures using physicochemical properties and spectroscopic methods [63]. Zhao et al. analyzed the dried latex of *E. resinifera* (Urumqi, China) using NOE experiments and 2D NMR spectroscopy and found 18 structurally diverse diterpenoids. This study found a new rhamnofolane diterpenoid (euphorblin O), 14 new ingol-type diterpenoids (euphorblins A-N), and 3 known analogues [25]. After one year, Wang and investigators completed their research and isolated euphorol J (a new euphane triterpene hydroperoxide) and a new rearranged tirucallane triterpene (euphorol K) in addition to a known compound (kansuinone) [60]. In the same year, euphoresin A and euphoresin B, two new ingol-type diterpenes, were isolated from Euphorbium methanol extract [61]. Qi et al. isolated euphatexol A and euphatexol B (two unusual euphane triterpenoids) from *E. resinifera* latex and elucidated their structures based on various spectroscopic data (1D, 2D NMR, and HRMS) [58]. In a recent study, Zhao et al. isolated 10 new nortriterpenes (euphorbiumrins A–J) from the same part of *E. resinifera* (Urumqi, China) using extensive spectroscopic analyses (HRESIMS, IR, 1D, 2D NMR, and UV) [64]. In the same year, two new 12-Deoxy-16-hydroxyphorbol 13,16-diesters were isolated from *E. resinifera* latex (Beni Mellal, Khenifera province, Morocco) using UHPLC−HRMS [21]. Moreover, Ourhzif et al. isolated euphorbioside monohydrate (Euphorbioside C) from the latex of this plant (Azilal, Morocco) using a single-crystal X-ray structure, NMR, LC/HRMS, and IR spectra [59]. In addition, Li et al. identified five new tritepenoids named as euphatexols (C–G) from the ethanol extract of Euphorbium (Xinjiang, China) using a group of spectroscopic methods (IR, HRMS, 1D, and 2D NMR data) [52]. The same authors isolated 3β-hydroxy-12α-methoxy-24-methylene-lanost-7,9(11)-dien; 3β-hydroxy-12α-methoxylanosta-7,9(11),24-triene; 3,7-dioxo-24-methylene-lanost-8-en; and 3,7-dioxo-lanosta-8,24-diene (new lanostane triterpenoids) from the latex of *E. resinifera* using several chromatography methods [51]. Furthermore, seven triterpenoids have been identified and isolated from *E. resinifera* ethanol extract. From this extract, they identified cycloartan-1,24-diene-3-one, cycloartan-1,24-diene-3-ol, 3β-hydroxy-lanosta-8,24-diene-11-one, lnonotusane C, eupha-8,24-diene-3β-ol-7,11-dione, eupha-24-methylene-8-ene-3β-ol-7,11-dione, and eupha-8,24-diene-3β,11β-diol-7-one) using several methods including ODS column chromatography, semi-preparative HPLC, and normal-phase silica gel column chromatography [53].

### 3.7.2. Honey Composition

For a long time, honey has been used traditionally not only as a sugar but also as a medicine to treat different diseases, including cataracts and ulcers, and in burn and wound healing. Researchers from various scientific fields have been studying the physicochemical properties (ash, pH, electrical conductivity, free lactone, water activity and moisture, and total acidity) and contamination of honey (heavy metal and pesticides) [36,65,66].

In 2005, Malika et al. (2005) conducted a study on a total of three samples of Moroccan *Euphorbia* honey from different botanical origins to evaluate their physicochemical characteristics including the ashes, pH, electrical conductivity, free lactone, water activity and moisture, and total acidity [65]. The microbiological properties were studied using standard plate count (SPC), fungi (yeasts and molds), bacteria sporulated, and total coliforms. The results showed a water activity of 0.55. The lactone acidity varied from 4.1 to 9 meq/kg, the free acidity varied from 29 to 33 meq/kg, and the total acidity showed an average of 37.53 meq/kg. An average pH of 4.27 was recorded, while the ash content was 0.5189 g/kg, the moisture was about 20%, and the conductivity was 0.7163 ms/cm. For the microbiological characteristics, the SPC varied from 20 to 200 cfu/g, a low levels of fungi (yeasts and molds) were reported, and no total coliforms were detected in the three samples of *E. resinifera* honey [65]. In the same context, Moujanni et al. (2017) studied the physicochemical characteristics of 29 samples of *E. resinifera* honey (Beni Mellal and Azilal provinces, Morocco) [1]. Most honeys showed good ripeness due to their low water content. The authors reported that average values of conductivity, ash, total acidity, and hydroxymethylfurfural were 451 μS/cm, 1.6 g/kg, 8 meq/kg, and 3.6 mg/Kg, respectively. Mineral investigation showed a presence of Mg, Ca, P, Na, Si, and S in honey in varying amounts with a high abundance of K element.

Resiniferatoxin

12,20-dideoxy-phorbol-13-isobutyrate

Euphorblin A

Euphol

**Figure 3.** *Cont.*

Euphorbol

Euphatexol A

Euphorol A

Euphorol B

Inoterpene B

Dammarenediol II

**Figure 3.** *Cont*.

Euphoresin A

Euphoresin B

Euphorbiumrin A

Euphorbiumrin C

**Figure 3.** Chemical structure of terpenoids identified in *E. resinifera* latex.

Recently, a study conducted by Terrab et al. (2021) investigated palynological and geographical characterization of 29 samples of *E. resinifera* honey distributed in a Protected Geographical Indication (PGI) area of the Middle Atlas Mountains (Morocco) [36]. Analysis of quantitative pollen of honey showed low levels of pollen (average number of pollen grains (NPG) = 1490/g honey), while the number of honeydew elements were in a low quantity ranging from 0 to 95/g of honey.

Bees collect pollen, honeydew, honey, and water from the environment. These last items contain different chemical contaminants and natural products, which are found in products consumed by humans and, therefore, which may pose a risk to public health. The honey produced could be considered as a bio-indicator of environmental pollution. In this context, Moujanni et al. (2017) investigated heavy metals presence (Cd, Hg, and Pb), and 202 pesticides residues including 53 organophosphorus (OPPs), 16 synthetic pyrethroïds (PYR), and 39 organochlorine (OCPs) using different methods of identification and quantification, including liquid chromatography–tandem mass spectrometry (LC-MS/MS), cold vapor mode hydride generation atomic absorption spectrophotometry (CV-AAS), graphite furnace atomic absorption spectrophotometry (GF-AAS), gas chromatography–flame photometric detector (GC-FPD), and gas chromatography–electron capture detector (GC-ECD) [66]. The authors of this study showed that the pesticide and heavy metal residue levels in all other samples were below the pre-set limits.

### 3.8. Pharmacological Investigation

*E. resinifera* is an important traditional medicinal species used by the Moroccan population against different disorders. Several others described in the bibliography numerous pharmacological activities, especially as antimicrobial, antioxidant, anti-inflammatory, antileishmanial, antitumoral, antitrypanosomal, immunomodulatory, and insecticide activities (Figure 4).

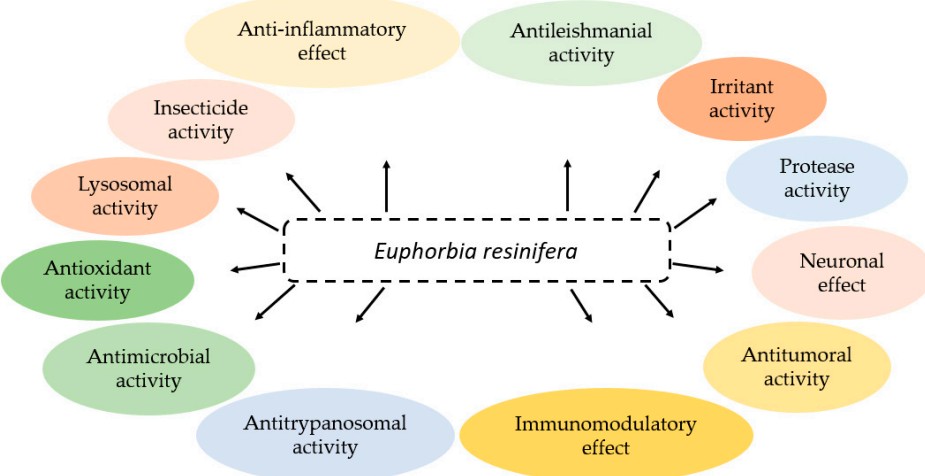

**Figure 4.** Biological properties of *E. resinifera*.

### 3.8.1. Antimicrobial Activity
Antibacterial Activity

Numerous research studies have evaluated the antibacterial activity of various parts of *E. resinifera* extracts and the antibacterial quality of *E. resinifera* honey. Several part extracts including roots, latex, aerial parts, flowers, and stems extracts were evaluated. Moreover, the most used method to evaluate the antibacterial activity was the microdilution assay. As reported, the antibacterial activity of different parts of *E. resinifera* extracts, type of extract, used method, tested strains, and key results are listed in Table 3.

**Table 3.** Antibacterial effect of *E. resinifera*.

| Use Part | Extracts | Used Method | Tested Strains | Key Results | References |
|---|---|---|---|---|---|
| Aerial parts | Aqueous extract | Agar direct contact method | *S. aureus* ATCC 25923 | - | [3] |
| | | | *P. aeruginosa* ATCC 27853 | - | |
| | | | *E. coli* ATCC 25922 | + | |
| Roots | Acetone extract | Microdilution technique | *E. coli* ATCC 35210 | MIC = $0.5 \pm 0.0$ mg/mL<br>MBC = $0.5 \pm 0.2$ mg/mL | [17] |
| | | | *S. aureus* ATCC 29213 | MIC = $0.5 \pm 0.0$ mg/mL<br>MBC = $1.0 \pm 0.2$ mg/mL | |
| | | | *S. typhimurium* ATCC 13311 | MIC = $1.0 \pm 0.1$ mg/mL<br>MBC = $1.5 \pm 0.2$ mg/mL | |
| | | | *B. subtilis* ATCC 10907 | MIC = $1.0 \pm 0.2$ mg/mL<br>MBC = $1.5 \pm 0.0$ mg/mL | |
| | | | *S. epidermidis* ATCC 12228 | MIC = $0.5 \pm 0.0$ mg/mL<br>MBC = $1.0 \pm 0.2$ mg/mL | |

**Table 3.** *Cont.*

| Use Part | Extracts | Used Method | Tested Strains | Key Results | References |
|---|---|---|---|---|---|
| Roots | Ethanol extract | Microdilution technique | *E. coli* ATCC 35210 | MIC = 0.3 ± 0.0 mg/mL<br>MBC = 0.5 ± 0.0 mg/mL | [17] |
| | | | *S. aureus* ATCC 29213 | MIC = 0.5 ± 0.0 mg/mL<br>MBC = 1.0 ± 0.2 mg/mL | |
| | | | *S. typhimurium* ATCC 13311 | MIC = 0.5 ± 0.0 mg/mL<br>MBC = 1.0 ± 0.2 mg/mL | |
| | | | *B. subtilis* ATCC 10907 | MIC = 1.0 ± 0.2 mg/mL<br>MBC = 1.0 ± 0.0 mg/mL | |
| | | | *S. epidermidis* ATCC 12228 | MIC = 0.5 ± 0.0 mg/mL<br>MBC = 1.0 ± 0.2 mg/mL | |
| | Ethyl acetate | Microdilution technique | *E. coli* ATCC 35210 | MIC = 0.4 ± 0.0 mg/mL<br>MBC = 0.3 ± 0.0 mg/mL | |
| | | | *S. aureus* ATCC 29213 | MIC = 0.2 ± 0.0 mg/mL<br>MBC = 0.3 ± 0.2 mg/mL | |
| | | | *S. typhimurium* ATCC 13311 | MIC = 0.5 ± 0.0 mg/mL<br>MBC = 1.0 ± 0.2 mg/mL | |
| | | | *B. subtilis* ATCC 10907 | MIC = 0.3 ± 0.0 mg/mL<br>MBC = 0.1 ± 0.0 mg/mL | |
| | | | *S. epidermidis* ATCC 12228 | MIC = 0.2 ± 0.1 mg/mL<br>MBC = 0.6 ± 0.2 mg/mL | |
| | Dichloromethane extract | Microdilution technique | *E. coli* ATCC 35210 | MIC = 0.5 ± 0.0 mg/mL<br>MBC = 1.0 ± 0.2 mg/mL | |
| | | | *S. aureus* ATCC 29213 | MIC = 0.5 ± 0.0 mg/mL<br>MBC = 1.0 ± 0.2 mg/mL | |
| | | | *S. typhimurium* ATCC 13311 | MIC = 1.0 ± 0.2 mg/mL<br>MBC = 0.5 ± 0.3 mg/mL | |
| | | | *B. subtilis* ATCC 10907 | MIC = 0.5 ± 0.0 mg/mL<br>MBC = 1.0 ± 0.2 mg/mL | |
| | | | *S. epidermidis* ATCC 12228 | MIC = 0.5 ± 0.0 mg/mL<br>MBC = 1.0 ± 0.2 mg/mL | |
| Roots | Methanolic extract<br>———<br>Ethyl acetate extract | Microdilution technique | *E. coli* ATCC 35210 | MIC ≥ 16 ± 0 mg/mL<br>MBC ≥ 16 ± 0 mg/mL | [17] |
| | | | *S. aureus* ATCC 29213 | MIC = 2 ± 0 mg/mL<br>MBC = 2 ± 0 mg/mL | |
| | | | *S. typhimurium* ATCC 13311 | MIC = 0.5 ± 0 mg/mL<br>MBC = 0.5 ± 0 mg/mL | |
| | | | *B. subtilis* ATCC 10907 | MIC = 0.5 ± 0 mg/mL<br>MBC = 0.5 ± 0 mg/mL | |
| Stems | Methanolic extract<br>———<br>Ethyl acetate extract | Microdilution technique | *S. aureus* ATCC 29 213 | MIC ≥ 16 ± 0 mg/mL<br>MBC ≥ 16 ± 0 mg/mL | [46] |
| | | | *B. subtilis* ATCC 3366 | MIC = 4 ± 0 mg/mL<br>MBC = 4 ± 0 mg/mL | |
| | | | *S. aureus* ATCC 29 213 | MIC = 2 ± 0 mg/mL<br>MBC = 2 ± 0 mg/mL | |
| | | | *B. subtilis* ATCC 3366 | MIC = 4 ± 0 mg/mL<br>MBC = 2 ± 0 mg/mL | |

**Table 3.** *Cont.*

| Use Part | Extracts | Used Method | Tested Strains | Key Results | References |
|---|---|---|---|---|---|
| Flowers | Methanolic extract | Microdilution technique | *S. aureus* ATCC 29 213 | MIC = 2 ± 0 mg/mL<br>MBC = 2 ± 0 mg/mL | [46] |
| | | | *B. subtilis* ATCC 3366 | MIC = 4 ± 0 mg/mL<br>MBC = 4 ± 0 mg/mL | |
| | Ethyl acetate extract | | *S. aureus* ATCC 29 213 | MIC = 4 ± 0 mg/mL<br>MBC = 4 ± 0 mg/mL | |
| | | | *B. subtilis* ATCC 3366 | MIC = 4 ± 0 mg/mL<br>MBC = 4 ± 0 mg/mL | |
| Latex | Isolated euphorbioside | Serial dilution methode | *E. coli* ATCC 25922<br>*S.aureus* ATCC 19433<br>*P. aeruginosa* ATCC 27853<br>*B. subtilis* ATCC 6633 | | [62] |
| Aerial parts | Hexanic extract | Well agar diffusion method | *Rhodococcus equi* | Φ= 18 mm | [17] |
| | Dichloromethane extract | Well agar diffusion method | *Rhodococcus equi*<br>*Rhodococcus sp GK1* | Φ = 18 mm<br>Φ = 15 mm | |
| Honey | - | Well agar diffusion and dilution range | *S. aureus* ATCC 6538<br>*E. coli* ATCC 10536 | Φ = 25.98 ± 0.11 mm<br>Φ = 13.84 ± 1.10 mm | [19] |

-: resistance, +: inhibition, Φ: diameter of inhibition.

Benmahdi and coworkers (2013) revealed that the antibacterial activity of aqueous extract of aerial parts of *E. resinifera* from Naama (Algeria) was observed only against *E. coli* strain [3]. One year later, Farah and coworkers (2014) studied the antibacterial effect of acetonic, ethyl acetate, dichloromethane, and ethanol extracts obtained from the roots of *E. resinifera* collected from Damnat (Morocco) against five strains: *Escherichia coli, Staphylococcus aureus, Staphylococcus typhimurium, Staphylococcus epidermidis,* and *Bacillus subtilis* [17]. They revealed that the most resistant species was *S. typhimurium*, with a MICs of 1.5–2.0 mg/mL and a MBCs of 2.0–2.5 mg/mL. They also reported that *E. coli* was the most sensitive (MICs of 1.0–1.5 mg/mL, and MBCs of 1.5–2.0 mg/mL). Moreover, they reported that the ethyl acetate extract was the most effective. In another research study investigated by Farah et al. (2014), they studied the antibacterial effect of polar extracts of roots, flowers, and stems of *E. resinifera* from Beni Mellal (Morocco) against *S. aureus, B. subtilis, Micrococcus luteus, Pseudomonas aeruginosa,* and *E. coli* [46]. Form this study, the authors revealed that ethyl acetate and methanolic extracts of all parts of this plant were active against *S. aureus* and *B. subtilis*. However, only the ethyl acetate extract revealed activity against *M. luteus* strains. In addition, they noticed that roots polar extracts of *E. resinifera* potentially can be designed to be more active than other parts. In addition, the antibacterial effect of a molecule isolated from the latex of *E. resinifera* was evaluated by Farah et al. (2014) [62]. The antibacterial effect of euphorbioside (from fresh latex of *E. resinifera*) and semi-synthetic euphorbioside was tested against *E. coli* and *S. aureus*. From this research, the authors demonstrated that numerous degrees of inhibition against the tested strains were observed for synthesized compounds. However, they noticed that the natural euphorbioside is completely inactive against the tested strains. Recently, according to Talbaoui et al. (2020), a dichloromethane extract of aerial parts of *E. resinifera* showed an inhibitory zone of 15 mm (at concentration 50 µg/mL) against *Rhodococcus sp GK1*, and both dichloromethane and hexane extracts present a moderate inhibitory zone of 18 mm for *R. equi* [18].

Honey presents the most energy-dense food in nature, which contains a high source of sugar (fructose and glucose), presents traces of several essential vitamins and minerals [67], and presents remarkable therapeutic effects [68]. Many Mediterranean countries

used *Euphorbia* honey to treat several diseases such as circulatory disorders, metabolic disorders, and digestive diseases [69]. In this context, Bendjamaa et al. (2020) evaluated the antibacterial effect of 37 *E. resinifera* honeys against *S. aureus* and *E. coli* using the well agar diffusion assay [19]. From this study, they found that all samples of honey inhibited the growth of bacteria at dilutions of 50% (*v/v*) whit 25.98 ± 0.11 mm, 13.84 ± 1.10 mm and as the highest inhibition zone for *S. aureus* and *E. coli*, respectively. Regarding the honey of *E. resinifera*, Bendjamaa et al. (2020) suggested that *E. resinifera* honey presents an opportunity as a natural adjunct to treat many diseases related to pathogens *E. coli* and *S. aureus*. In addition, the authors suggested that honey might act through pH, honey's phytochemical characteristics, viscosity, and content of $H_2O_2$ [20]. Moreover, Moujanni et al. (2017) evaluated the compliance of *E. resinifera* honey with bacteriological recommendations and showed an absence of coliforms (total and fecal coliforms), *Clostridium perfringens*, and sporus of *Bacillus cereus*, *Shigella* spp., and *Salmonella* spp. [70].

Different factors (extract used, used part, localization of plant, the experimental used, extraction methods, and bioactive compounds present in the plant) explain the variation of the results in these studies [71–73]. All these in vitro studies reported that *E. resinifera* presented a considerable antibacterial effect. However, further in vivo and clinical studies on this effect are needed.

Antifungal Activity

The antifungal activity of *E. resinifera* extract has been described by only one study, which was conducted by Benmehdi et al. (2013) against two fungal strains. Table 4 summarizes the antifungal effect of this endemic Moroccan plant [3]. In 2013, Benmehdi et al. (2013) evaluated the antifungal effect of *E. resinifera* aqueous extract in terms of percentage of radial growth on solid medium [3]. They proved that this extract inhibits the mycelium growth of two fungal organisms, *Aspergillus flavus* and *Penicillium expansum*, which are harmful to humans at various concentrations. The results showed that the mycelial growth inhibition rates ranged from 64.14% to 85.51% for *A. flavus* and from 60.14% to 85.51% for *P. expansum*. On the one hand, the potent antifungal activity of *E. resinifera* extract underscores the effect of bioactive compounds; therefore, other research should be conducted to reveal the effect of chemical classes against various fungal strains. On the other hand, there is a lack of research on extracts against other fungal pathogen strains. Thus, investigation of extracts, chemical classes, and/or honey of *E. resinifera* should be explored against various pathogenic fungal strains to prevent contamination. In addition, further studies should be conducted in depth (in vitro, in vivo, and in vivo preclinical tests) to develop an antifungal drug.

**Table 4.** Antifungal activity of *E. resinifera*.

| Use Part | Extracts Used | Used Method | Test Strains | Key Results | References |
|----------|---------------|-------------|--------------|-------------|------------|
| Aerial parts | Aqueous extract | Growth radial technique on solid medium | *A. Flavus* MTTC 2799 *P. expansum* MTTC 1344 | I = 64.14% to 85.51% I = 60.14% to 85.51% | [3] |

3.8.2. Antioxidant Activity

Numerous studies were conducted to search for natural antioxidants from *E. resinifera*. Many investigators were interested in the evaluation of the antioxidant activity of several extracts of different parts of *E. resinifera* [3,17]. All these studies were conducted using DPPH, superoxide, and nitric oxide assays.

Table 5 regroups all research carried out regarding the antioxidant effect of *E. resinifera* including type of extract, parts used, methods used, and key results. Benmehdi et al. (2013) investigated the anti-radical activity of methanol, flavonoids, and alkaloids extracts of *E. resinifera* aerial parts (from Algeria) [3]. The authors of this study showed that the methanol, flavonoids, and alkaloids extracts exhibited an antioxidant activity with a value of $IC_{50}$ = 0.0086 mg/mL, $IC_{50}$ = 0.378 mg/mL, and $IC_{50}$ =1.171 mg/mL, respectively. The

antioxidants' effect using DPPH was shown by their ability to donate hydrogen [3]. In 2014, Hanane and coworkers (2014) followed the same method to show the antioxidant effect of *E. resinifera* root extracts collected in Morocco [17]. They studied the anti-radical activity of the root extract of this plant using a DPPH scavenging assay. Ethyl acetate extract expressed the strongest anti-radical activity with a $SC_{50}$ value of $18.20 \pm 0.41$ μg/mL. In addition, acetone, ethanol, and dichloromethane extracts showed moderate effects ($SC_{50}$ = 98.44 μg/mL, 65.01 μg/mL, and 122.15 μg/mL, respectively). In the same year, Hanane et al. (2014) tested the anti-DPPH activity of *E. resinifera* roots. From this study, a higher antioxidant activity of roots methanolic extract was reported compared with roots ethyl acetate extract (10.01 μg/mL and $18.85 \pm 0.12$ μg/mL, respectively). Furthermore, the anti-radical effect of roots methanolic extract was significantly higher than those achieved from stems or leaves extracts. Furthermore, Hanane et al. (2014) proved a correlation between the anti-DPPH activity and the presence of phenolic compounds [17]. Recently, Boutoub and colleagues (2021) assessed the antioxidant power of aqueous extract of *E. resinifera* aerial parts and honey of *E. resinifera* collected from Moroccan honey using DPPH assay, superoxide, and nitric oxide methods [17]. This study showed that the honey sample of *E. resinifera* had an anti-DPPH effect with $IC_{50}$ = $80.1 \pm 1.1$ mg/mL, a superoxide activity with $IC_{50}$ = $3.70 \pm 0.0$ mg/mL, and a nitric oxide radical effect with $IC_{50}$ = $88.2 \pm 0.8$ mg/mL. Indeed, various in vitro research investigations have shown that the variation of scavenging activity can be attributed to several parameters including climatic conditions, botanic origin, composition of phenolic compounds, and type of extracts [20,74]. Furthermore, the synergistic effects between the minor and the main biological compounds and the evaluation of other parts of *E. resinifera*, as well as latex, should be taken into consideration.

**Table 5.** Antioxidant activity of *E. resinifera*.

| Used Part | Extracts | Used Method | Key Results | References |
|---|---|---|---|---|
| Aerial parts | Methanolic extract | DPPH | $IC_{50}$ = 0.0086 mg/mL | [3] |
| | Flavonoids extract | | $IC_{50}$ = 0.378 mg/mL | |
| | Alkaloids extract | | $IC_{50}$ = 1.171 mg/mL | |
| Aerial parts | Aqueous extract | DPPH | $IC_{50}$ = 0.370 mg/mL | [17] |
| Honey | - | DPPH | $IC_{50}$ = $80.1 \pm 1.1$ mg/mL | [17] |
| | | Superoxide | $IC_{50}$ = $3.70 \pm 0.0$ mg/mL | |
| | | Nitric oxide | $IC_{50}$ = $88.2 \pm 0.8$ mg/mL | |
| Roots | Dichloromethane extract | DPPH | $SC_{50}$ = $122.15 \pm 0.52$ μg/mL | [17] |
| | Ethyl acetate extract | | $SC_{50}$ = $18.20 \pm 0.41$ μg/mL | |
| | Ethanol extract | | $SC_{50}$ = $65.01 \pm 0.32$ μg/mL | |
| | Acetone extract | | $SC_{50}$ = $98.44 \pm 0.13$ μg/mL | |
| Roots | Methanol extract | DPPH | $IC_{50}$ = $10.01 \pm 0.17$ μg/mL | [46] |
| | Ethyl acetate extract | | $IC_{50}$ = $18.85 \pm 0.12$ μg/mL | |

$IC_{50}$: inhibitory concentration 50%; $SC_{50}$: scavenging concentration 50%.

### 3.8.3. Antitumoral Activity

Only one study investigated the antitumoral effect of the aerial parts of *E. resinifera* [18]. Table 6 lists the studies that investigated the anticancer properties of *E. resinifera* including the extract tested, the part used, the cell lines, and the main results. In 2020, Talbaoui and colleagues (2020) tested the effect of dichloromethane and methanol extracts of *E. resinifera* (Morocco) against cancer cell lines, namely kidney adenocarcinoma of hamster (BSR), monkey kidney cancerous cell lines (Vero), and embryonal rhabdomyosarcoma cancerous cell lines (RD) using the MTT assay [18]. The results showed that the dichloromethane and

methanol extracts of *E. resinifera* have significant antitumor effect against BSR cell lines with a value of IC$_{50}$ = 77.2 μg/mL. The anticancer effects of *E. resinifera* were attributable to the main compounds in the extracts. However, minor compounds can also participate in the anticancer activity through their synergistic effects and/or through their additive effects with the major compounds [18,72].

**Table 6.** Antitumoral activity of *E. resinifera*.

| Part Used | Extracts | Cells Lines | Key Lines | References |
|---|---|---|---|---|
| Aerial parts | Hexane extract | Animal cells Vero | IC$_{50}$= 266.43 μg/mL | [18] |
| | | Human cells RD | IC$_{50}$= 50.7 μg/mL | |
| | Dichloromethane extract | Animal cells BSR | IC$_{50}$= 77.2 μg/mL | |
| | | Vero | IC$_{50}$= 79.2 μg/mL | |
| | Methanolic extract | Human cells RD | IC$_{50}$= 67.57 μg/mL | |
| | | Animal cells BSR | IC$_{50}$= 200 μg/mL | |

BSR: kidney adenocarcinoma of hamster; RD: monkey kidney cancerous cell lines; Vero: embryonal rhabdomyosarcoma cancerous cell lines.

It has been reported above that *E. resinifera* is mainly used traditionally to heal cancer. However, there is a lack of research that investigates the anticancer properties of different parts of this plant. In this context, further investigations should be conducted to explore the effect of *E. resinifera* on different types of cancer.

3.8.4. Anti-Inflammatory Effect

In the last years, the free radical nitric oxide (NO) was proved to be associated with a potent cytotoxic effect responsible in pathogenesis of different human diseases. In this context, researchers oriented their studies to neutralize NO [75]. Five triterpenoids isolated and identified from *E. resinifera* latex (from Xinjiang, China) were investigated by Li et al. (2021) and tested for their anti-inflammatory properties (Table 7) [52]. The anti-inflammatory effect was carried out in vitro using nitric oxide (NO) production inhibition assay induced by lipopolysaccharide (LPS) in RAW264.7 cells. It has reported as result that euphatexols C–G showed a moderate inhibitory effect on the production. The euphatexol E has the ability to inhibit the production of NO with an IC$_{50}$ value of 21.89 μM, which is near to the positive control used (dexamethasone with an IC$_{50}$ = 20.35 μM) [52]. These compounds showed a potent anti-inflammatory effect. However, further in vivo tests should be undertaken to evaluate the efficacy and safety of these compounds.

**Table 7.** Anti-inflammatory effect of triterpenoids isolated from *E. resinifera*.

| Part Used | Compounds Isolated | Experimental Approach | Key Lines | References |
|---|---|---|---|---|
| Latex | Euphatexol C | NO * production inhibition assay | IC$_{50}$ = 22.30 μM | [52] |
| | Euphatexol D | | IC$_{50}$ = 48,04 μM | |
| | Euphatexol E | | IC$_{50}$ = 21.89 μM | |
| | Euphatexol F | | IC$_{50}$ = 38.15 μM | |
| | Euphatexol G | | IC$_{50}$ = 41.15 μM | |

* NO: nitric oxide.

3.8.5. Antileishmanial Activity

Leishmaniasis is an endemic disease caused by a protozoan parasite. This contain over 20 different species of Leishmania parasites. A wide range of sandfly species are known to transmit Leishmania parasites. Visceral, cutaneous, and mucocutaneous leishmaniasis are the three main forms of the disease. The third most common parasitic cause

of death is due by visceral leishmaniasis. For therapeutic options, there are a limited number of available drugs to treat this disease due to their side effects, as well as drug resistance [76]. In fact, during the last decade, natural compounds and their derivatives have been identified as a source of therapeutic agents. It was found that medicinal plants were effective against leishmaniasis [77]. The antileishmanial activities, on *Leishmania infantum* promastigotes, of 12 semisynthetic terpenoid derivatives obtained by several chemical modifications of *E. resinifera* major compounds (Demnat, Morocco) were studied by Mazoir et al. (2011) [20]. These triterpenes at different concentrations (100, 10, and 1 µg/mL), extracted from the latex with methanol, showed significant antiparasitic effects on *Leishmania infantum* with an $ED_{50} < 10$ µg/mL (Table 8). 3β-Tosyloxy-24-methylen-elemo-lanosta-8,24-dien-11-one (C38H53O4S) and 3β-Tosyloxy-24-methylen-elemo-lanosta-8,24-dien-7,11-dione (C38H54O3S) were the most cytotoxic to *L. infantum* ($ED_{50} = 2.13$ µg/mL and $ED_{50} = 5.16$ µg/mL, respectively) with less cytotoxic effect to mammalian Chinese hamster ovary cells ($ED_{50} > 100$ µg/mL). Since Leishmania parasites have a strict requirement for specific endogenous sterols for survival and growth, Mazoir et al. (2011) suggested that the triterpenes used can interfere with sterol metabolism [20].

### 3.8.6. Antitrypanosomal Activity

Chagas, an endemic disease, is caused by the parasite *Trypanosoma cruzi*, especially in Latin America. Undiagnosed Chagas disease can damage the heart; consequently, it is a cause of sudden cardiac deaths. Nifurtimox and benznidazole are fairly effective drugs utilized in the acute stage of this disease. However, in the developed stage, such drugs lack the desired safety and efficacy profiles. The emergence of drug-resistant strains highlights the interest in discovering new drugs, especially from natural sources [78]. The *Euphorbia* genus has been reported to have antitrypanosomal activity due to its wide variety of terpenoids. In fact, the *in vitro* activity on Trypanosoma cruzi epimastigotes of 12 semisynthetic terpenoid derivatives obtained by chemical modifications of *E. resinifera* major compounds (Demnat, Morocco) were studied by Mazoir and coworkers (2011) [20]. These triterpenes at different concentrations (100, 10, and 1 µg/mL), extracted from the latex with methanol, showed significant antitrypanosomal effects on *T. cruzi* with a value of $ED_{50} < 10$ µg/mL (Table 8). 3β-Tosyloxy-24-methylen-elemo-lanosta-8,24-dien-11-one ($C_{38}H_{53}O_4S$) and 3β-Tosyloxy-24-methylen-elemo-lanosta-8,24-dien-7,11-dione ($C_{38}H_{54}O_3S$) were the most cytotoxic to *T. cruzi* ($ED_{50} = 5$ µg/mL and $ED_{50} = 6.98$ µg/mL, respectively) with less cytotoxic effect to mammalian Chinese hamster ovary cells ($ED_{50} > 100$ µg/mL). The mechanism by which these compounds exhibited their cytotoxic activity was not fully understood and the authors suggested that the triterpenes used can interfere with sterol metabolism [20]. Further studies are needed with natural triterpenes of the *Euphorbia* genus in order to discover new bioactive drugs against Chagas disease caused by protozoan parasites.

### 3.8.7. Protease Activity

Hemostasis is a physiological process, which consists of fibrinolysis (clot dissolution) and coagulation (fibrin clot formation) that stops bleeding from a blood vessel when an injury occurs. Cardiovascular diseases, resulting from an unbalance between fibrinolysis and coagulation systems, are the major causes of mortality and morbidity worldwide. Thrombolytics or fibrinolytics are therapeutic drugs used to treat thrombosis. These drugs have common adverse effects (viz. hemorrhage) and a short half-life in human blood circulation. Indeed, new natural antithrombotics offer a promising therapeutic alternative [79].

One of the main target sources of antithrombotic agents is plant proteases due to their ability to interfere with antiplatelet activity and blood coagulation. In fact, the effect of the purified protease from *E. resinifera* latex on the human platelet function and blood coagulation system was investigated by Siritapetawee et al. (2020) [80] (Table 8). In order to investigate the effects of blood-circulating protease inhibitors on fibrinogenolytic activity, the activity of purified protease was not inhibited by antithrombin III and α2-

macroglobulin, naturally circulating inhibitors. Regarding fibrinolytic activity, the in vitro effect of purified protease on human fibrin clots was investigated using SDS-PAGE under reducing conditions. Accordingly, the hydrolysis of all chains of the fibrin clot was reported by the purified protease at a concentration $\geq 0.5$ μM. In addition, the in vitro anticoagulant activity of purified protease was also evaluated using activated partial thromboplastin time (APTT) and prothrombin time (PT) assays. The purified enzyme showed an anticoagulant effect as demonstrated by prolonged induction in both APTT and PT. Furthermore, using ADP-induced light transmittance aggregometry in a microwell pate, the effect of purified protease on human platelet aggregation was investigated. The purified protease decreases the aggregation activity of platelets in a dose-dependent manner, whereas at concentrations $\geq 1$ μM of enzyme, platelet function was abolished. In parallel, cytotoxicity assessment of the purified protease was also evaluated. Thus, the purified protease was found to be non-toxic against human peripheral blood mononuclear cells and four common blood groups of human red blood cells (A, B, O, and AB) (all Rh+) [80]. From these results, the purified protease from *E. resinifera* latex can have a potential role to be used to treat thrombosis. Similarly, in a previous study, serine protease from *E. resinifera* latex was purified by Siritapetawee et al. (2019) and their biochemical properties against fibrinogenolytic activity were investigated using human fibrinogen as a substrate [80]. From this study, serine protease cleaved human fibrinogen with optimal conditions at pH 5.0 and 45 °C; it also exhibited a wide pH stability range of 1–14 and showed a tolerance to denaturation at a temperature of 65–66 °C. The purified protease hydrolyzed fibrinogen with a catalytic efficiency ($V_{max}/K_m$) of 116.8 ng/μM/min, a Michaelis constant ($K_m$) of 4.95 μM, and a maximal velocity ($V_{max}$) of 578.1 ng/min. However, purified protease from *E. resinifera* latex showed lower catalytic efficiency for cleaving human fibrinogen compared to human plasmin, suggesting the potential use of this protease in medical protection and treatment against thrombosis.

### 3.8.8. Anti-Xanthine Oxidase, Antilipoxygenase, Antiacetylcholinesterase, and Antityrosinase Activities

Studying the inhibition-important enzymes is an effective therapeutic strategy. Xanthine oxidase, lipoxygenase, acetylcholinesterase, and tyrosinase are examples of a few enzymes that are pharmacologically and physiologically important. The primary target by the inhibiting of different enzymes is estimated by nearly 47% of the total available drugs. Thus, the discovery of new and potent enzyme inhibitors is a crucial area of interest in pharmaceutical and pharmacological research [81]. In fact, the inhibitory effect of a wide range of enzymes has been shown by the genus *Euphorbia*. Boutoub et al. (2021) compared the inhibition of lipoxygenase, acetylcholinesterase, xanthine oxidase, and tyrosinase enzyme effects of aqueous extracts and honeys of *E. resinifera* [17]. On the one hand, the capacity of *E. resinifera* aqueous extracts was tested for the inhibition of four enzymatic activities. The aqueous extract at different ratios (1:20, 1:50, and 1:100) was obtained after 2 h of extraction, and the aqueous extract with a 1:100 ratio had significantly better activity to inhibit xanthine oxidase than the other ratios ($IC_{50} = 26.1$ mg/mL). The same ratio from the aqueous extract of *E. resinifera* after 1 h of extraction showed the best antilipoxygenase activity ($IC_{50} = 0.99$ mg/mL). Independent of extraction time or the ratio, it has been shown that the aqueous extract of *E. resinifera* exhibited better activities than *Euphorbia officinarum*. For the tyrosinase activity, the aqueous extracts with a ratio 1:100 at 30 min of extraction exhibited better activity ($IC_{50} = 1.49$ mg/mL). On the other hand, the inhibitory activities of *E. resinifera* honey on xanthine oxidase ($IC_{50} = 71.7$ mg/mL), lipoxygenase ($IC50 = 32.7$ mg/mL), acetylcholinesterase ($IC_{50} = 44.7$ mg/mL), and tyrosinase ($IC_{50} = 11.5$ mg/mL) was lower compared to that of *E. officinarum* honey, which exhibited higher $IC_{50}$ values of 94.9 mg/mL, 46.8 mg/mL, 165 mg/mL, and 54.9 mg/mL, respectively [17]. The aqueous extracts showed systematically higher in vitro activities compared with honey samples. These things considered, aqueous extracts and honeys of *E.*

*officinarum* and *E. resinifera* showed their potential as a potent enzyme inhibitory that can be used in therapeutic agent to treat numerous diseases.

### 3.8.9. Neuronal Effect

Neurological disorders, vascular disorders, and traumatic injuries cause brain damage and irreversible neuronal loss, which leads to motor dysfunction, cognitive impairment, and personality changes. The brain has the ability to generate new neurons from neural stem cells (NSCs) in the subventricular zone (SVZ) and dentate gyrus (DG) of the hippocampus; however, rarely does neuronal regeneration occur in damaged brain regions. Hence, there is the need to look for molecules potentiating the ability of NSC to produce new neurons in damaged brain regions, facilitating functional recovery [82].

The vanilloid receptor subtype 1 (VR1) is a sensor for noxious stimuli on the peripheral terminals of sensory neurons. It is expressed in various brain nuclei as well as non-neuronal tissues, placing it in a broader clinical context than pain perception, and increasing the potential for unforeseen side effects, particularly after exposure to term to vanilloid [83]. This receptor VR1 is controlled by capsaicin and noxious heat, and its activation leads to the release of pro-inflammatory peptides that bind to the calcitonin gene (CGRP), leading to hyperalgesia and peripheral sensitization [84]. Potent VR1-blocking agonists, such as resiniferatoxin and capsaicin extracted from the latex of *E. resinifera*, generate peripheral analgesia via ablation of VR1-expressing nociceptors and desensitization of VR1. They have beneficial effects for detrusor hyperreflexia and overactive bladder, where the afferent branch of the voiding reflex is affected [84]. A study conducted by researchers at the National Institutes of Health (NIH) Clinical Center showed that resiniferatoxin extracted from *E. resinifera* (Morocco) reduced post-surgical pain in rats. According to the NIH, resiniferatoxin is chemically similar to capsaicin but is 500 times more potent and has the ability to blocks pain-related signaling in nerve cells (Kuehn, 2018). Kuehn (2018) reported that injection of resiniferatoxin in rats with incised paws decreases pain-related behaviours in resiniferatoxin-treated rats compared to untreated animals [27]. Recently, several derivatives of phorbol (polyhydroxylated tetracyclic diterpene, present as esters in Euphorbiaceae) were showed to be able to induce neural progenitor cell (NPC) proliferation through activation of protein kinase C (PKC) and stimulation of neurogenesis in (SVZ) and (DG) and particularly, 12-deoxyphorbol (DPB) isolated from *E. resinifera*. This exhibited the highest ability to stimulate neurogenesis in vivo by long-term intranasal administration. Thus, they showed that mice treated with this compound improved memory and learning by facilitating the release of TGFα and promoting the proliferation of (NSC) [21]. In 2021, Ezzanad et al. (2021) isolated and identified two new 12-deoxy-16-hydroxyphorbol 13,16-diesters from *E. resinifera* latex [21]. From this study, the authors showed an ability of these compounds to stimulate the release of TGFα in a classical PKC-dependent manner thanks to the C-12 and C-13 or C-13 and C-16 hydroxyl groups which must be esterified so that these compounds have this ability, thereby increasing NPC proliferation and stimulating neurogenesis. Opioid-based narcotics play an important role in pain management, but they can have central-nervous-system-mediated negative side effects. The improvement of pain management can be enhanced through the discovery of effective non-opioid alternative treatments and by targeting the peripheral nervous system to prevent side effects.

### 3.8.10. Immunomodulatory Effect

The immunomodulatory activity of *E. resinifera* was investigated in vitro by Kreher et al. (1990) [85]. From the compounds identified by HPLC and MS from *E. resinifera* latex, 11 triterpenes, 3 12-deoxyphorbol esters, and 2 ingol esters were identified. Only the 12-deoxyphorbol esters induced lymphocyte proliferation in vitro, while triterpenes and ingol esters showed no immunomodulating properties. In 2019, the immunomodulatory effect of *E. resinifera* was also studied in vivo by Issiki et al. (2019) using young adult male mice [31]. The aqueous extract of aerial parts of *E. resinifera* (Beni Mellal, Morocco) was given at 5 g/kg body weight (single oral dose) for acute toxicity tests, whereas the aqueous

extract of *E. resinifera* was administrated daily and orally by gavage to different groups at different concentrations (0.1, 0.5, 1.0, 2.5, or 5 g/kg body weight) during a period of 28 days for sub-acute toxicity tests. The level of the delayed-type hypersensitivity (DTH) reaction and the evaluation of antibodies production was evaluated to test the immunological activity. The result of this research showed a significant increase in antibody titer values in the group treated without a toxic dose (1 g/kg) compared to the control. In addition, at 1 g/kg, *E. resinifera* aqueous extract increased the DTH reactivity response compared to control animals (37% and 25%, respectively). These results showed that *E. resinifera* aqueous extract potentiates cellular immunity as well as humoral immunity [31]. The high source of terpenoids in this plant may act as an immunomodulatory and could justify the high number of lymphocytes infiltrates found in tissues examined.

### 3.8.11. Irritant Activity

Irritant activity of *E. resinifera* resin and latex was investigated in vivo on the ear of a mouse by Hergenhahn (1984) [50]. Esters of the polyfunctional diterpenes, namely 12-deoxyphorbol, ingenol, 12-deoxy-16-hydroxyphorbol, 12,20-dideoxyphorbol, resiniferonol, and ingol, were purified and chemically characterized from latex and resin. The irritant effect of these compounds was estimated as irritation units. Two esters of ingol did not showed irritant effect on the mouse ear [50]. Similarly, two 12,20-dideoxyphorbol esters were also practically nonirritant. However, ingenol, 12-deoxy-16-hydroxyphorbol, 12-deoxyphorbol, and resiniferonol all presented less or more irritant activity. Similarly, Zayed et al. (1984) also isolated two ester compounds (13,20-diesters and 13-monoesters) from resin of *E. resinifera* and investigated in vivo their irritant activity on the ear of a mouse [23]. The 13,20-diesters generally have a lower irritant activity than the corresponding 13-monoesters. The authors suggested that the length of the acyl chain of the 13-monoesters strongly affects the irritant activity.

### 3.8.12. Lysosomal Activity

Lysosomal biosynthesis induction activity of euphorblin B and euphorblin D isolated from *E. resinifera* latex was studied in vitro using the HeLa cell line [25]. To investigate whether euphorblin B and euphorblin D possess the ability to induce lysosomal biosynthesis, different concentrations of these compounds (10, 20, 40, and 60 μM) were tested for their activity at 1, 3, 6, and 9 h [25]. LysoTracker Red staining was conducted to examine lysosome induction. Most of the isolated compounds increased the LysoTracker staining intensity. HeLa cells were treated with different concentrations during a period of 3 h, and the compounds used for the lysosome induction were increased in a concentration-dependent manner, with the greatest increase observed at 60 μM. Likewise, using 40 μM of test compounds at different times (0, 1, 3, 6, and 9 h), a lysosomal biosynthesis induction effect has been reported in a time-dependent manner. Thus, the isolated diterpenoids from *E. resinifera* latex showed notable abilities to induce lysosome biosynthesis at different concentrations and time [25].

### 3.8.13. Toxic Effect

The aqueous and organic extract of *E. resinifera* have been used for numerous pharmacological properties. However, few studies investigated the toxicity effect of this plant. Since ancient times, *E. resinifera* latex has been used in traditional folk medicine to treat many medicinal indications. However, latex presents frequent intoxications, which restricted its use for external mediation and veterinary medicines. In this context, Hergenhahn et al. (1984) isolated, identified, and analyzed diterpene esters responsible for tumor promoters in tumor-promoting experiments [50]. From this study, the authors found that mixed homologous stimulator RL 13 is equivalent in use to the standard tumor promoter 12-0-tetradecanoylphorbol-13-acetate (TPA), but at 10 times the dose of TPA. They also reported that the long-chain ingenol-3-esters and the 12-deoxyphorbol-13-esters exhibited a moderate promoting effect, and the short-chain 12-deoxyphorbol and 12-deoxy-

16-hydroxyphorbol esters ascribed a cytotoxic effect. In the same year, regarding the tumor-promoting activity of 13-monoesters, three synthesized 12-deoxyphorbol-13- acetate and benzoate, Zayed et al. (1984) investigated their tumor-promoting effect and found that they were not active at the doses tested (both tested at $p = 40$ nmol), and only the latter showed tumor-promoting activity [23]. At selected doses, tetradecanoate ($p = 80$ nmol) and octadecanoate ($p = 100$ nmol) appeared to be equivalent to TPA ($p = 5$ nmol). It should be noted that 13-tetradecanoate ($p = 12.5$ nmol) was previously found to be comparably as effective as TPA ($p = 10$ nmol). In 2004, Therre and collaborators (2004) investigated the DNA-damaging and chemoprotective effects of *E. resinifera* [86]. The complex mixtures (ethanolic extracts according to HAB 2000) were carried out with the help of two standardised genotoxicity assays on metabolically competent, human hepatoma cells (HepG2). The in vitro micronucleus test was used for investigation for direct genotoxic effects. To determine anti- or cogenotoxic effects, the plant extracts were compared to the effects of the standard mutagen benzo[a]pyrene in the single-cell gel electrophoresis (comet assay). *E. resinifera* showed strong cytotoxic effects up to dilution D4. The dilutions D3 and D4 also showed dose-dependent cogenotoxic effects. In addition, Therre and coworkers (2004) investigated the cogenotoxic effects of two medicinal products (Euphorbium compositum nasal drops SN®, Euphorbium compositum SN®), as well as three Euphorbia diterpene esters (13-O-isobutyryl-12-deoxyphorbol-20-acetate, 13-O-(phenylacetyl)-12-deoxyphorbol-20-acetate, and 12-O-tetradecanoylphorbol-13-acetate) [86]. They showed that neither the medicinal products nor the diterpene esters were relevant in cogenotoxic effects. Only 12-O-tetradecanoylphorbol-13-acetate caused significant amplification of benzo[a]pyrene-induced DNA damage, but this was the quantitative effect of *E. resinifera* extract that cannot be explained. Recently, Li et al. (2021) investigated the cytotoxicity effect of euphatexols C–G (five new triterpenoids) isolated from *E. resinifera* latex on Cell Counting Kit-8 [52]. At 10 µL, all compounds showed a non-cytotoxic effect on cells.

The toxic effect of *E. resinifera* was studied in vivo using young adult male mice [31]. The aqueous extract of *E. resinifera* (Beni Mellal, Morocco) was given orally in a single oral dose (5 g/kg body weight) for the acute toxicity tests. For sub-acute toxicity tests, the aqueous extract of *E. resinifera* was given orally by gavage daily (0.1, 0.5, 1.0, 2.5, or 5 g/kg body weight) during a period of 28 days. Thereafter, biochemical analysis as well as histopathological examination of liver, kidney, and spleen were evaluated [31]. Regarding biochemical analysis and at a dose of 0.1 and 0.5 g/kg, indicators of kidney activity (urea and creatinine) did not demonstrate any differences among treated and control groups. However, especially at the higher dose of 5 g/kg, significant dose-dependent increases in ALAT and ASAT (principal indicators of liver toxicity) were observed. To confirm the results of biochemical analysis, histopathological examination showed no detectable alteration in spleen tissues or kidney between the treated mice with a concentration of 2.5 g/kg and the controls. However, at a dose of 5 g/kg, high interstitial inflammatory lymphocytes infiltrates and accentuated hemosiderin were recorded in kidney and spleen tissues, respectively. For the liver tissue, at 0.5 and 1 g/kg, some low hepatocytes lesions were observed, whereas at 5 g/kg, a massive steatosis centrilobular microvacuolar with nucleus duplication of the hepatocyte was observed. From these studies, it can be shown that the toxicological activity of *E. resinifera* is rarely studied and additional preclinical tests (especially in vivo tests) are mandatory to determine the safe dose.

### 3.8.14. Insecticide Activity

Various researchers oriented their research by the use of triterpenes derivatives compounds to treat the insect pest. This is the case of Smaili et al. (2018) who worked on the effects of four semi-synthesized triterpenes derivatives from the Moroccan *E. resinifera* latex, namely 3b-tosyloxy-4a, 14a-dimethyl-5a-cholesta-7,9-diene; 4α, 14α-Dimethyl-5α-cholesta-7,9-dien-3b-ol; 24-methylene-elemolanosta-8,24-dien-3-one; and elemolanost-8-en-3,11,24-trione [20,87]. From this study, the four semi-synthesized triterpenes completely inhibited sprouting at concentrations of 500 mg/mL and 100 mg/mL. In addition, the effect

of inhibiting sprouting was less affected when the concentration dropped to 50 mg/mL and 10 mg/mL. Furthermore, the tomato seeds germinated in the presence of four triterpenes derivatives induced resistance to Verticillium wilt caused by Verticillium dahliae. Concentrations as low as 10 mg/mL were effective in reducing the external symptoms of wilt of Verticillium manifested by leaf lesion and growth restriction, as well as internal symptoms such as vascular discoloration (Table 8).

**Table 8.** Other activities of *E. resinifera*.

| Activities | Use Part | Extracts | Experimental Approach | Key Results | References |
|---|---|---|---|---|---|
| Antileishmanial activity | Latex | Methanol extract | MTT assays against *Leishmania infantum* (strain PB75) | $ED_{50} < 10$ μg/mL | [20] |
| Antitrypanosomal activity | Latex | Methanol extract | MTT assays against *Trypanosoma cruzi* (strain Y) | $ED_{50} < 10$ μg/mL | [20] |
| Protease activity | Latex | Crude extract | Activity of purified protease against blood-circulating protease inhibitors | No inhibition of protease activity | [80] |
| | | | In vitro effect of purified enzyme on human fibrin clots using SDS-PAGE under reducing conditions | Hydrolysis of all chains of the fibrin clot by purified protease with concentrations $\geq 0.5$ μM | |
| | | | In vitro anticoagulant activity of purified protease using PT and APTT assays | APTT and PT prolongation inducted in both in the presence of purified protease | |
| | | | Activity of purified protease against human platelet aggregation using light transmission aggregometry | At concentrations $\geq 1$ μM, purified protease abolishes platelet function | |
| | Latex | Crude extract | Optimum pH: activity of purified protease was assayed at different pH values (1–14) | Optimum pH = pH 5.0 | [80] |
| | | | Optimum temperature: activity of purified protease was assayed at temperature between 15 and 95 °C | Optimum temperature = 45 °C | |
| | | | pH stability: purified protease incubated at different pH (1–14) at 37 °C during a period of 24 h | Purified protease was stable in broad pH range (1–14) | |
| | | | Temperature stability: purified protease was incubated at temperature between 15 and 95 °C for 2 h | Purified protease was stable at temperatures up to 65–66 °C | |
| | | | Kinetic studies: $K_m$, $V_{max}$, and $V_{max}/K_m$ of the fibrinogenolytic activity of purified protease determined by incubating 2 μg of the enzyme with various amounts of human fibrinogen (0–32 μM) | $K_m = 4.95$ μM; $V_{max} = 578.1$ ng/min; $V_{max}/K_m = 116.8$ ng μM$^{-1}$ min$^{-1}$ | |
| Enzyme inhibitory potential | Aerial parts and honey | Aqueous extracts and honey | Percentage of inhibition of lipoxygenase, xanthine oxidase, tyrosinase, and acetylcholinesterase calculated by the determination of $IC_{50}$ | Higher in vitro activities of aqueous extracts observed than in the respective honey samples | [17] |

**Table 8.** *Cont.*

| Activities | Use Part | Extracts | Experimental Approach | Key Results | References |
|---|---|---|---|---|---|
| Immunomodulatory effect | Latex | Crude extract | In vitro studies of 12-deoxyphorbol esters, ingol esters, and triterpenes | Only the 12-deoxyphorbol esters showed a stimulation of lymphocytes proliferation | [85] |
| | Aerial parts | Aqueous extract | In vivo acute toxicity (at dose 5 g/kg body weight) and sub-acute toxicity tests (at dose 0.1, 0.5, 1.0, 2.5, or 5 g/kg) | A significant increase in antibody titer values in the group treated without toxic dose (1 g/kg) compared to the control Increase in the DTH reactivity response (37%) compared to control animals (25%) at a dose of 1g/kg | [31] |
| Irritant activity | Latex and resin | Methanol extract | Irritant activity of purified compounds of *E. resinifera* resin and latex was investigated in vivo on the ear of a mouse; irritant activity of these compounds was estimated as irritation units | Two esters of ingol did not exhibit irritant activity Two 12,20-dideoxyphorbol esters were also practically nonirritant Esters of 12-deoxyphorbol, 12-deoxy-16-hydroxyphorbol, resiniferonol, and ingenol showed less or more irritant activity | [50] |
| | Resin | Methanol extract | Irritant activity of isolated 13,20-diesters and 13-monoesters from resin of *E. resinifera* was investigated in vivo on the ear of a mouse; irritant activity of these compounds was estimated as irritation units | The 13,20-diesters generally have a lower irritant activity than the corresponding 13-monoesters | [23] |
| Liver effect | Propolis | Oil extract | Activities of the SGPT and SGOT were analyzed before, during, and after 10 months of the beginning of the treatment | A significant reduction of SGPT and SGOT concentrations of all patients | [24] |
| Lysosomal activity | Latex | Methanol extract | Different concentrations of the isolated diterpenoids (10, 20, 40, and 60 μM) were tested for the lysosomal effect at different time intervals (1, 3, 6, and 9 h) using HeLa cell line using LysoTracker Red staining to observe the induction of lysosome | The isolated diterpenoids showed notable abilities to induce lysosome biosynthesis in concentrations and time-dependent manner | [25] |
| Skin effect | Resin | Alcohol extract | Sixty human patients with verrucae plantares treated with alcohol-resin solution (95–30%) on the central keratotic area of the verruca | After 96 h, a complete removal of the central keratotic area of the verrucae was observed | [26] |

**Table 8.** *Cont.*

| Activities | Use Part | Extracts | Experimental Approach | Key Results | References |
|---|---|---|---|---|---|
| Toxic effect | Aerial parts | Aqueous extract | In vivo acute toxicity (at dose 5 g/kg body weight) and sub-acute toxicity tests (at dose 0.1, 0.5, 1.0, 2.5, or 5 g/kg) during a period of 28 days; biochemical analysis (creatine, urea, ALAT and ASAT) and histopathological examination of liver, kidney, and spleen were investigated | At the doses of 0.1 and 0.5 g/kg, creatinine and urea did not show any differences A significant dose-dependent increase in ASAT and ALAT At dose lower than 2.5 g/kg, no detectable alteration in spleen tissues or kidney At 5 g/kg, alteration was reported in kidney and spleen tissues At doses of 0.5 and 1 g/kg, some low hepatocytes lesions were observed Using 5 g/kg, massive hepatocytes lesions were observed | [31] |

### 3.8.15. Clinical Trials

Due to the encouraging results from in vivo and in vitro investigations, various patient clinical trials were tested to explore the safety and clinical activity of *E. resinifera* compounds or/and extracts on numerous diseases including asthma, cervical ripening, induction of labor, verrucae, corneal edema associated with systemic dopaminergic agents, and pain relief [26,28,30,88].

In 1953, Goldblum and Curtis (1953) investigated the use of 30% alcoholic solution of *E. resinifera* to treat 60 patients affected with verrucae plantares. This solution was used to the central keratotic area of the verruca and protected with adhesive tape to avoid Euphorbium from spreading onto normal tissue [26]. After 48 h of application, the wart was re-clipped with a scalpel and Euphorbium was reapplied. As a result, the sixty patients with verrucae plantares were treated, with complete disappearance of the verrucae in all but two patients after a four-month follow-up period. From this study, the authors suggested that lysing the collagen of the dermis and lysing action on the cells of the rete mucosum are the mechanism of pathways by which Euphorbium exhibited the disappearance of the verrucae (Goldblum and Curtis, 1953) [26]. However, further study on the effect of terpenoids compounds to treat verrucae should be taken into consideration.

For a clinical trial of asthma studies, Zissu et al. (2011) investigated the case report of a patient with a post-influenza asthma attack, eight days previous, with new symptoms: nose and greenish mucus from the nasal passages, candidiasis from his mother and sister, and jealousy with violence against the latter [30]. The patient was immediately controlled by Antimonium tartaricum. In addition to the basic weekly treatments (Psorinum and sulfur), this patient received *E. resinifera* 7CH once daily. There was an improvement after the second consultation and no respiratory allergies were observed (no asthma and no high fever).

Another clinical study assessing the effectiveness of the herb Unani formulation for cervical maturation and labor induction was conducted by Sultana and investigators (2014) [88]. They reported the effect this polyherb Unani on 38 pregnant women with a gestation period of 38–42 weeks. The patients received orally the polyherb Unani composed of *Cinnamon tamara* 3 g, *Peganum harmara* 5 g, *Pinus longifolia* 1 g, and *Gentiana lutea* 1 g

powder up to 4 times at 6 h intervals. In addition, they received a pessary placed in the vagina containing *E. resinifera* 0.5 g, *Gossypium herbaceaum* 2 g, and borax 3 g (up to 4 times at 6 h intervals). From this study, the authors showed that 12.3 ± 4.7 h was the mean interval from induction to delivery. Of the 38 pregnant women, 84.2% delivered spontaneously, 7.8% delivered vaginally after injection of oxytocin and/or cerviprime, and 7.8% delivered by caesarean, with no babies admitted to the neonatal ward. In addition, *E. resinifera* demonstrated antibacterial and antioxidant properties, and the polyherbal formulation was beneficial for cervical ripening and labor induction with no adverse effects [88].

The effect of propolis honey-oil extract from *E. resinifera* was investigated on serum concentrations of glutamate transaminases in chronic hepatitis C patients [24]. The oil extract was prepared using olive oil; then, the oil extract was mixed with multifloral honey (15%) for the non-diabetic patients and with *E. resinifera* honey (15%) for the diabetic ones. Propolis honey-oil preparation was administrated orally three times one hour before the meal. Thereafter, the activities of two glutamate transaminases, serum glutamate oxaloac-etate transaminase (SGOT), and serum glutamate-pyruvate transaminase (SGPT) were analyzed before, during, and after 10 months of the beginning of the treatment. Propolis honey-oil extract showed obvious improvement in normal liver function. SGPT and SGOT concentrations of all patients showed a significant reduction during the administration period (from 190 IU/L to 90 IU/L, and from 200 IU/L to 80 IU/L, respectively). In parallel, a significant inhibition of fatigue symptoms in all patients including those who had been resistant to the conventional therapy was observed. Propolis honey-oil extract from *E. resinifera* showed an improved effect on normal liver function and supported the beneficial effect of apitherapy in patients with chronic hepatitis [24].

As for the pain relief investigation, Kuehn (2018) noted that chemicals from *E. resinifera* cactus plants from Morocco adequately reduce postoperative pain in rats in research from the National Institutes of Health (NIH) Clinical Center [27]. Thus, a phase I clinical trial at NIH Clinical Center was conducted to evaluated the effect of resiniferatoxin injectable in the central nervous system. From this study, the pain was related and nerve cell signaling was unaffected during training.

Recently, Mancera and Wadia (2019) aimed to describe the development and resolution of corneal edema in a patient using *E. resinifera* resin [28]. After discontinuation of the interfering drug and treatment of exposure to resiniferatoxin, corneal edema gradually resolved and visual acuity returned to baseline.

## 4. Conclusions

In this bibliometric study, we described the ethnomedicinal use, taxonomy, phytochemistry, pharmacology, and clinical trials of *E. resinifera*. Our analysis of numerous published studies revealed that the latex is the main part used in Moroccan traditional folk medicine to treat different diseases, especially cancer. In the phytochemical analysis of *E. resinifera*, terpenoids were identified as the main chemical compounds in the latex extract. The variation of *E. resinifera* terpenoids at different phonological stages is an important parameter. Therefore, it will be interesting to study this variation to determine the main secondary metabolites in the developmental stage. Meantime, the pharmacological effects of *E. resinifera* extract and compounds have been studied in vitro and in vivo, including antioxidant, antimicrobial, anti-inflammatory, antileishmanial, immunomodulatory, and antitrypanosomal activities. However, despite the traditional use of *E. resinifera* to treat cancer, only one pharmacological study investigated its antitumoral effect. Therefore, more anticancer studies should be conducted on various types of cancer to explore the potential of this plant. In addition, studying the pharmacokinetic and pharmacodynamics properties of bioactive compounds of *E. resinifera* is necessary to guide clinical medication and ensure the stability of drug treatment in different pathologies. Moreover, despite the pharmacological studies of *E. resinifera* extract and compounds, little literature data investigated the toxicological effects of this plant. Thus, a deep toxicological investigation is mandatory to

guarantee its safety. This review can help researchers to grasp the current state of research on *E. resinifera* and orient investigators for further research.

**Author Contributions:** A.K. conceived the idea, collected the data, provided the guidance, and reviewed the article, wrote a part of the manuscript and structured the article as per the journal guidelines. A.C. wrote a part of the manuscript and reviewed the final version of the article. O.H., K.B., and S.K. wrote a part of the manuscript. A.R. wrote a part of the manuscript and drew the chemical structures using ChemDraw Ultra 12.0 software. F.Z. organized the data and cited the references as per the journal guidelines, and wrote a part of the manuscript. H.R., R.N., E.M. and M.M. reviewed and corrected the manuscript. All authors have read and agreed to the published version of the manuscript.

**Funding:** This research received no external funding.

**Institutional Review Board Statement:** Not applicable.

**Data Availability Statement:** Not applicable.

**Conflicts of Interest:** The authors declare no conflict of interest.

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
