# Peer review of "Ethnomedicinal Use, Phytochemistry, Pharmacology, and Toxicology of Euphorbia resinifera O. Berg. (B): A Review"

_2673-5636, doi:10.3390/jzbg4020029_

Round 1

Reviewer 1 Report

The manuscript is descriptive and well written. The tables and figures are well described. The discussion is complete. The manuscript in an up to date review. 

Author Response

Thank you

Reviewer 2 Report

The manuscript entitled ‘Ethnomedicinal Use, Phytochemistry, Pharmacology, and Toxicology of Euphorbia resinifera O. Berg. (B): A Review’ is a nice review highlighting the significance of Euphorbia resinifera. The manuscript is well written and has a scope for publication. I have some suggestions;

Botanical names should be always in italics; check Line 29 32; 96.

As the species is endemic, I suggest that information regarding its population status, possible threats and conservation should be added. By adding such information, it will be a comprehensive review and will better suit the journal scope.

Further, information on its cultivation, tissue culture approaches and conservation polices should be added, if relevant.

Author Response

All remarks have been corrected. Thank you.

Reviewer 3 Report

The review article titled "Ethnomedicinal Use, Phytochemistry, Pharmacology, and Toxicology of Euphorbia resiniferaO. Berg. (B): A Review" by Hmidouche et al. provides a comprehensive overview of the ethnobotanical uses, phytochemical composition, pharmacological activities, and toxicological effects of Euphorbia resinifera O. Berg. This Moroccan endemic plant is known for its various medicinal properties and has been used traditionally to treat a wide range of diseases. The authors have collected and compiled information from various sources, including scientific journals and ethnobotanical studies, to present a detailed account of the plant's traditional use, chemical composition, and therapeutic potential. However, some issues must be addressed before further action.

1.     The body of this review article is written like an original article which must be changed, like materials and methods, results, etc. This applies to abstract also. 

2.     AbstractHere are some grammatical and scientific errors I found in the abstract:

·      Always write the names of the plants in italics. Follow this throughout the article. 

·      In the first sentence, "different types of cancer" should be changed to "various types of cancer" for better grammar.

·      In the second sentence, "previous reports on E. resinifera, concerning its botanical description, taxonomy, geographical distribution, medicinal use, bioactive compounds, toxicology, and pharmacological effects" is a bit long and could be rephrased for better readability.

·      In the third sentence, "Different scientific search engines including Scopus, PubMed, Web of Science, SpringerLink, SciFinder, Wiley Online, and Google Scholar were consulted to collect data about E. resinifera" could be rephrased as "We searched various scientific databases, such as Scopus, PubMed, Web of Science, and others, to collect data on E. resinifera."

·      In the fourth sentence, "antitumoral" should be changed to "antitumor" for better grammar.

·      In the fifth sentence, "skin inflammation" should be changed to "inflammatory skin conditions" for better clarity.

·      In the seventh sentence, "immuno-modulatory" should be changed to "immunomodulatory" for better grammar.

·      In the eighth sentence, "These biological activities, based on evidence, support the use of E. resinifera in folk medicine" is a bit vague and could be rephrased for better clarity.

·      In the ninth sentence, "in-deep" should be changed to "in-depth" for better grammar.

·      Overall, the abstract could benefit from rephrasing to improve grammar and clarity.

3.     Introduction: 

Grammatical mistakes:

·       "which present" should be "which presents" (line 53).

·       "hypoglycemic" should be "and hypoglycemia" (line 59).

·       "This specie" should be "This species" (line 59).

·       "used traditionally against different illnesses" should be "traditionally used against different illnesses" (line 62).

·       This promotes us to right should be This promotes us to write” (line 77).

·        

Scientific mistakes:

·       The statement "E. resinifera is used traditionally as an abortifacient" is concerning and should be clarified or removed as it can be harmful to pregnant women.

·       The phrase "the in vivo toxicological effects of E. resinifera extracts were rarely studies" should be "the in vivo toxicological effects of E. resinifera extracts have been rarely studied".

·       The statement "no review was published to summarize all these studies of this Moroccan endemic plant, E. resinifera" is not accurate as there are already some review articles published on this plant.

4.     All figures must have the proper cited references and copyright permissions. 

5.     Taxonomy and geographical distribution:

There are a few grammatical and scientific errors in the paragraph:

·      In the first sentence, "Euphorbusresin" should be "Euphorbia resin" and "dicotyledonous" should be "dicotyledon".

·      In the second sentence, "under-branched" should be "under-branching".

·      In the fifth sentence, "with, the main" should be "with the main".

·      In the seventh sentence, "appears" should be "appear".

·      In the tenth sentence, “Thus, it extends in the central valley of the High Atlas in small 124 isolated units” should be “It extends into the central valley of the High Atlas in 124 small, isolated units”.

·      In the twelfth sentence, "with temperate, cool, and exceptionally cold variants" should be "including temperate, cool, and exceptionally cold variants."

6.     Conclusion: 

There are a few grammatical errors in the text. Here is a corrected version:

·      In this bibliometric study, we report on the ethnomedicinal use, taxonomy, phytochemistry, pharmacology, and clinical trials of E. resinifera. Our analysis of numerous published studies revealed that the latex is the main part used in Moroccan traditional folk medicine to treat different diseases, especially cancer. In the phytochemical analysis of E. resinifera, terpenoids were identified as the main chemical compounds in the latex extract. The variation of E. resinifera terpenoids at different phonological stages is an important parameter. Therefore, it will be interesting to study this variation to determine the main secondary metabolites in the developmental stage. Meantime, the pharmacological effects of E. resinifera extract and compounds have been studied in vitro and in vivo, including antioxidant, antimicrobial, anti-inflammatory, antileishmanial, immunomodulatory, and antitrypanosomal activities. However, despite the traditional use of E. resinifera to treat cancer, only one pharmacological study investigated its antitumoral effect. Therefore, more anticancer studies should be conducted on various types of cancer to explore the potential of this plant. In addition, studying the pharmacokinetic and pharmacodynamic properties of the bioactive compounds of E. resinifera is necessary to guide clinical medication and ensure the stability of drug treatment in different pathologies. Moreover, despite the pharmacological studies of E. resinifera extract and compounds, little literature data have investigated the toxicological effect of this plant. Thus, a deep toxicological investigation is mandatory to guarantee its safety. This review can help researchers grasp the current state of research on E. resinifera and orient investigators for further research.

Author Response

(The authors gave the same response as above.)

Round 2

Reviewer 3 Report

Dear Authors,

I appreciate the significant improvements you have made to the manuscript. However, there are still a few changes required before it can be accepted for publication. Please address the following concerns:

  1. Chemical formulas: Please review and revise the chemical formulas throughout the manuscript, ensuring that they are written correctly and consistently. Proper formatting should be used, including subscript for numbers, capitalization for element symbols, and adherence to standard conventions.

  2. Uniformity of references in tables: Ensure that the references in the tables are presented uniformly. This includes using a consistent citation style, proper formatting, and maintaining the same order as they appear in the main text.

  3. Uniformity of units of analyses: Carefully examine the units of analysis used in tables and other parts of the manuscript, making sure they are consistently written and formatted. This involves using proper abbreviations, capitalization, and spacing, as well as employing the same units for the same types of measurements.

Once you have addressed these concerns, please resubmit the revised manuscript for further consideration. We look forward to reviewing your updated submission.

Author Response

  1. Some formulas have been corrected and underlined in purple. Thank you.
  1. We have checked that the references in the tables are presented uniformly (table 3 reference 20). Thank you.
  1. We have checked the Uniformity of units of analyses. Thank you.